# Assessment of the Quality of post-abortion care records in 20 Public Health Facilities in Uganda: What are the gaps and how can we improve quality?

Lynn Muhimbuura Atuyambe[1]*, Justine N. Bukenya[1], Samuel Etajak[2], Jesca Nsungwa-Sabiiti[3], Richard Mugahi[3], Paul Mbaka[3], Onikepe Owolabi[4], Sharon Kim-Gibbons[4], Kristy Friesen[4], Arthur Bagonza[1]

1 Department of Community Health and Behavioral Sciences, Makerere University, School of Public Health, Kampala, Uganda, 2 Department of Disease Control and Environmental Health, Makerere University, School of Public Health, Kampala, Uganda, 3 Ministry of Health, Kampala, Uganda, 4 Vital Strategies, New York, United States of America

* atuyambe@musph.ac.ug

## Abstract

### Background

Effective strategies for reducing maternal mortality depend on the accuracy of data. In sub-Saharan Africa, however, weak health systems often result in significant gaps in data collection and analysis. These gaps can lead to misreporting, missing data, or double-counting at health facilities, ultimately skewing the aggregated data reported by Ministries of Health. This is a big problem given that the data shapes planning and policy of the country. In this study, we assessed the data quality of Post-abortion Care (PAC) records across 20 health facilities in Uganda.

### Methods

A two-stage design was employed. In stage 1, DHIS2 data from 80 public health facilities were reviewed for completeness, timeliness, and internal consistency (2018–2021). In stage 2, 20 facilities that met a predefined eligibility criterion (≥ 50% discrepancy or missing data during stage 1) were purposively selected for in-depth validation. Data were collected retrospectively from facility registers for the last 7 months of 2021 and prospectively for the first 6 months of 2022. Data quality was assessed using adapted WHO data quality review (DQR) metrics.

### Results

Health facilities achieved a 100% timeliness rate for report submissions, with all documents provided by the 7th day of each month. Although a statistically significant positive correlation was observed between the number of women who received PAC

**Data availability statement:** All relevant data are within the manuscript and its Supporting Information files.

**Funding:** Finding for this work was secured from Vital Strategies Inc. New York USA. How ever the study ended a year a go. The funders had no role in study design, data collection and analysis, decision to publish, or preparation of the manuscript.

**Competing interests:** The authors have declared that no competing interests exist.

and those who received post-abortion family planning (PAFP) (r = 0.083, p < 0.001), the association was very weak suggesting limited internal consistency between the two indicators and raising questions about the reliability of data linkages between care and follow-up services in routine reporting systems. Moreover, comparing facility records with DHIS2 data revealed that outpatient records were more accurate than inpatient records, with PAC data being more inaccurate than other maternal health indicators. Further analysis by facility type indicated a higher number of case-load outliers at Health Centre IIIs and IIs (1049) compared to hospitals and Health Centre IVs, highlighting disparities in data quality across different levels of healthcare facilities.

## Conclusion:

Despite high submission rates, PAC data accuracy was notably low, highlighting the need for better data management and record-keeping, particularly in lower-level health facilities. Addressing these disparities is crucial for improving maternal health outcomes, emphasizing the necessity for targeted interventions to enhance data accuracy and reliability within Uganda's health system.

## Introduction

Global targets set under SDG3 aim at reducing the global maternal mortality ratio (MMR) to below 70 per 100,000 live births by 2030 [1]. Despite several interventions, limited data quality assessments have been conducted in areas such as maternal and reproductive health, often overlooking post-abortion care data entirely [2–4]. This is consistent with global findings that highlight persistent challenges in the quality, completeness, and use of routine health information systems in low-and middle-income countries, especially in sub-Saharan Africa [11,6]. In Uganda, the Ministry of Health and its partners advocate for comprehensive understanding and continuous monitoring of factors contributing to maternal morbidity and mortality by collecting high-quality data from healthcare facilities transmitted via the DHIS2 system. The current maternal mortality rate stands at 189 per 100,000 live births [7], with abortion contributing significantly to this figure. Recent data reveals that severe abortion morbidity ratio is 2,063 per 100,000 live births, with a near-miss ratio of 938 and a mortality ratio of 23 per 100,000 live births. The Eastern region faces the most severe outcomes, largely due to insufficient readiness of healthcare facilities to provide adequate postabortion care [8].

Despite several interventions, limited data quality assessments have been conducted in areas such as maternal and reproductive health, often overlooking post-abortion care data entirely. Notably, restrictive abortion laws in Uganda pose a significant challenge to collecting reliable PAC data [9,10]. Good quality post-abortion care (PAC) data plays a crucial role in tracking the prevalence and severity of admissions for abortion-related diagnoses within communities [11,12] and thus understanding and proposing strategies to prevent abortion-related maternal morbidity

and mortality [13]. However, in low and middle-income countries (LMIC) like Uganda, where routine health data from the health system is sparse and often of questionable quality, collecting high-quality PAC data remains highly challenging due to misinterpreted restrictive anti-abortion laws.

Although DHIS2 has been implemented at the district level to enhance data reporting and management, many lower-level health facilities rely on traditional paper-based systems. This dependence creates several challenges: paper-based methods are susceptible to errors, such as incomplete records and inaccuracies [14]. As a result, reports from these lower-level facilities often contain mistakes, compromising the data quality ultimately reported to district and national levels. This shift from paper to digital systems has been slowed by many health workers lacking computer literacy and limited training opportunities due to resource constraints. In addition to limited access to computers and poor internet connectivity, other barriers such as high patient workloads, inadequate dedicated data personnel, low motivation for accurate reporting, and weak supervision mechanisms further constrain data collection and reporting at lower-level health facilities [15–17]. By examining the quality of post-abortion care data within a constrained policy environment, this research aims at informing planning and budgeting for PAC services High quality DHIS2 data are essential for identifying service delivery gaps, forecasting resource needs, and guiding evidence-based decisions on staffing, medical supplies, and training, particularly for maternal health services that are often underfunded or overlooked.

The primary objective of this study was to evaluate the quality of routinely collected data documented in DHIS2 across 80 randomly selected health facilities in Uganda.

We utilized the World Health Organization's data quality review toolkit [18] to assess key dimensions of data quality, including completeness, timeliness and consistency.

This will eventually facilitate planning and decision-making based on improved data. The findings from this assessment provide insights into the status of PAC data across public health facilities, and will provide important evidence for policymakers to make decisions that address maternal morbidity and complications related to abortion. It also provides insight into aspects of how post-abortion care is managed in communities where comprehensive abortion care is restricted.

## Methods

### Ethical approval and consent to participate

Ethical clearance to conduct this assessment was sought from the Makerere University School of Public Health Research and Ethics Committee (SPHREC) reference number SPH-2022–226 as well as from the Uganda National Council of Science and Technology (UNCST) (HS2260ES). Informed consent was sought from participants involved in the study before the commencement of the assessment. The Ministry of Health (MoH), district health officers, and hospital review boards provided administrative clearance to assess the selected districts, cities, and health facilities. Initially, eight districts were selected based on PAC caseload stratification (four with high and four with low caseloads). However, during the study period, Gulu city and Mbarara city formed on July 1, 2020 had already been granted city status and administratively separated from Gulu district and Mbarara district, respectively. Given this change, we applied for amendment and received ethical approval for these two cities. Informed consent was sought from all health management information personnel and health facility in-charges. For medical records, our research teams had no access to information that could identify individual participants during or after data collection in this study.

### Study design and setting

This assessment used a two-stage sequential design. Stage 1 comprised secondary analysis of routine DHIS2 data from 80 sampled public facilities (2018–2021). Stage 2 comprised an on-site record audit in 20 of the 80 facilities that met a pre-specified eligibility criterion ≥ 50% of monthly PAC register entries missing or discrepant in stage 1 (Fig 1)

Uganda has a population of 45.9 million people with the majority being female [19]. According to the 2022 Demographic and Health Survey estimates, Uganda's MMR stood at 189/100,000 live births, showing a 44% decline from 336 in 2016

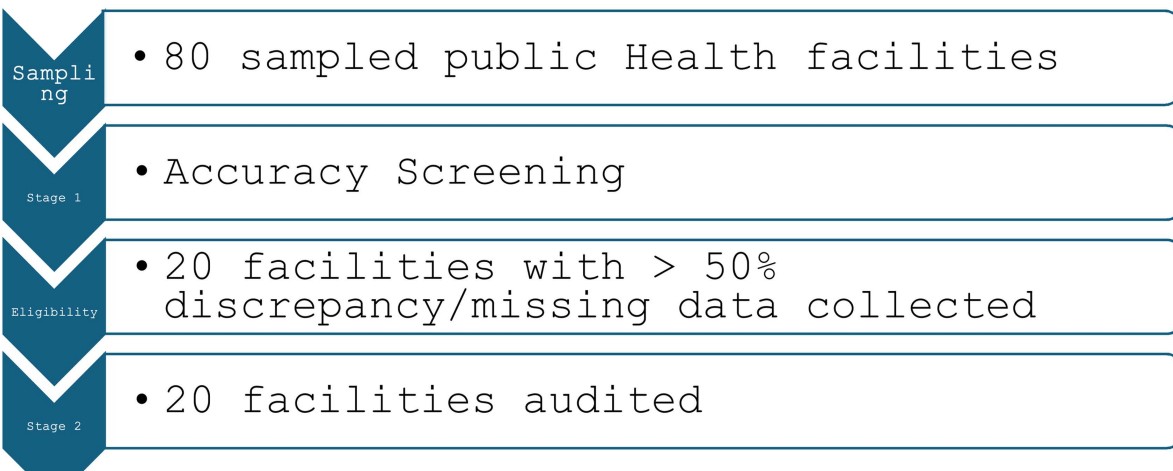

**Fig 1. Health facility selection process.**

Uganda has a total of 8,639 health facilities, 7,631 of which are predominantly public and provide Post-Abortion Care (PAC). The public health facilities are organized hierarchically. At the base are community health workers (CHWs) or volunteers. Above them are Health Centre IIs (HC IIs), which offer preventive, promotive, and outpatient curative health services, outreach care, and emergency services to a population of 5,000 people per center.

Next in the hierarchy are HC IIIs which provide preventive, promotive, outpatient curative, maternity, inpatient health services, and laboratory services for a catchment population of 20,000 people. Following HC IIIs are HC IVs serving 100,000 people with a broader range of services including emergency surgery, blood transfusion, and comprehensive laboratory services. District hospitals come next, covering a catchment of 500,000 people. They offer all the services HC IVs provide, along with in-service training, consultation, and research for the communities. Regional Referral Hospitals serve a population of 2,000,000 and provide specialized services such as psychiatry, ENT, ophthalmology, dentistry, intensive care, radiology, and pathology, in addition to the services offered by district hospitals.

At the top of the hierarchy are the National Referral Hospitals, which cater to a population of 10,000,000 people. These hospitals offer comprehensive specialist services and are heavily involved in teaching and research [20].

In Uganda, PAC is only provided by a gynaecologist or medical doctor who are mostly found at Health Centre IVs, district hospitals, regional referrals and National referrals. In some instances, however, doctors are found at Health Centre IIIs. With the advent of manual vacuum aspirators, the World Health Organization (WHO) recommends training of cadres such as Auxiliary nurses, nurses, midwives, and clinicians who can be found at lower-level health facilities to provide PAC. [21]

## Sample size calculation and sampling

**Stage 1.** We determined the sample size for facilities included in this assessment by obtaining the total number of health facilities that offer post-abortion care in Uganda. In total, 7,631 public health facilities out of the 8,639 health facilities were engaged in PAC. We then calculated the probability (p) of finding a health facility that offers PAC. This was calculated as (7,631/8,639*100) giving us a p = 0.88 using the Kish Leslie formula. We used the calculated prevalence (p), a margin of error of 1.96 at a 95% confidence level. We determined 162 health facilities as the total sample size of health facilities to be visited for the DQA.

However, since the number of health facilities offering PAC services is finite (the number is limited and known), the sample size was adjusted using the sample size calculation for finite populations. Ultimately, we ended up with 79 health

facilities. Due to proximity, we merged the Southern and Central regions, giving us four regions. With the four regions, we rounded off to 80 facilities, with 20 health facilities per region for better comparison.

From each of the four regions (Northern, Western, Eastern, and Central), we selected districts based on two criteria. The first criterion was districts with a high number of PAC cases (where high referred to more than 1000 cases per year). The second criterion was districts with low PAC cases (where low refers to less than 500 cases per year) as per the DHIS2 from January 2021 to December 2021. Districts with caseloads in between the high and low caseloads were not part of the study because the investigators wanted districts with clear-cut differences as far as caseloads were concerned. This was followed by a random selection of two districts per region meeting the above criteria. Districts meeting the criteria above had all health facilities offering PAC arranged in alphabetical order. Sampling proportionate to the health facility numbers then ensued. Districts selected as having high caseloads included: Gulu, Mbarara, Mityana, and Kamuli, while districts with low caseloads included: Nwoya, Kiruhura, Nakasongola, and Pallisa.

**Stage 2.** To validate the data from the initial assessment, which had significant gaps due to missing records at certain health facilities, a follow-up visit here referred to as Stage 2 in this study, was conducted among 20 health facilities. Only facilities with more than 50% discrepancy/missing data were included in the final analysis (n = 20) because we wanted to focus on validating and understanding the data quality challenges in the most affected facilities. Data for the last seven months of 2021 and the first six months of 2022 were collected. These months were purposefully selected because the financial year in Uganda begins in June, and the authors aimed at ruling out challenges associated with financial shortfalls that often occur before the start of the financial year.

## Data sources

For this assessment, data were obtained from two primary sources: facility-reported records within the District Health Information System 2 (DHIS2) and an independent facility survey. In the DHIS2, the only indicators available for Post-Abortion Care (PAC) were related to abortions due to Gender-Based Violence (GBV), abortions from other causes, and the number of post-abortion women who received family planning. However, we also reviewed additional indicators such as antenatal care (ANC) visits, total maternity ward admissions, total births, live births, and live births at discharge, as the same cadre of healthcare workers manages these. Our aim was to understand why there are discrepancies in the accuracy of some indicators, despite being documented by the same staff in health facilities.

A trained research team extracted data from the facilities' paper-based outpatient, maternity and delivery registers.

Our data collection started on October 1st 2022, and concurrently, investigators retrieved data from DHIS2. This data abstraction process, encompassing both the manual registers and DHIS2, spanned two months. Data was extracted from June 2021 to September 2022, after which a comparative analysis ensued between the data documented in the paper-based registers and the information extracted from DHIS2. Variables from which the data were extracted are shown in Table 1.

## Data quality assessment

The quality of DHIS2 data was evaluated based on three routine data quality dimensions defined by the World Health Organisation's data quality review toolkit: completeness and timeliness, internal consistency, and external consistency [18]. Table 2 details the evaluation criteria for each quality metric and the data sources utilized. In addition, a stratified analysis was performed according to the types of healthcare facilities.

## Data analysis

Internal consistency and outlier detection were assessed using the Median Absolute Deviation (MAD) method, following the Hampel X84 rule. Observations were flagged as outliers if their values lay beyond +5 times the MAD from the district-level or facility-type-level annual median. For indicators containing outliers, implausible values were replaced with the

**Table 1. Maternal and Newborn indicators used for comparison with PAC Data.**

| Maternal and Newborn Health data elements | Uganda's routine health information system | |
| --- | --- | --- |
| | Facility register | DHIS2 |
| Post Abortion Family Planning | X | X |
| **Abortions** | | |
| Gender-Based Violence (GBV) | X | X |
| Other causes | X | X |
| Incomplete evacuations | X | X |
| ANC 1st visit for women | X | X |
| ANC 4th visit for women | X | X |
| ANC 4 + visits for women | X | X |
| Total ANC contacts/visits (new clients + re-attendances) | X | X |
| **Admissions** | | |
| Deliveries in unit – Total | X | X |
| Deliveries in unit – Live births | X | X |
| Deliveries in unit – Fresh stillbirth – Total | X | X |
| Deliveries in unit – Macerated stillbirth – Total | X | X |
| Malaria in pregnancy (cases) | X | X |
| High blood pressure in pregnancy (cases) | X | X |

corresponding district-level annual median. Annual ratios were computed to evaluate post-abortion family planning (PAFP) uptake in relation to outpatient, inpatient, and total post-abortion care PAC) cases, as well as the ratio of first antenatal care (ANC) visits to four or more visits among women. To assess the strength and direction of association over time, pearson correlation coefficients and corresponding p-values were calculated for each indicator pair, with statistical significance set at $p < 0.05$. Additionally, simple linear regression models were fitted to explore the relationships and assess linear consistency between indicators, particularly examining the association between adjusted PAFP counts and adjusted PAC counts.

## Results

### Stage 1- Desk review of all 80 facilities (2018–2021)

The initial comparison of raw facility registers with DHIS2 data (2018–2021) from 80 health facilities showed low internal consistency accuracy scores: outpatient department (OPD) records reached a maximum of 13.3% in 2020, while inpatient records peaked at 10.7% accuracy in 2019. When considering OPD and inpatient records together, the overall accuracy was 10.5% in 2020. While the WHO DQR toolkit stipulates a 75% threshold for completeness of reporting, it provides no specific cutoff for accuracy. Therefore, these low percentages highlight substantial discrepancies between source registers and DHIS2 reports rather than non-compliance with a formal WHO accuracy benchmark. These low accuracy levels were linked to missing records in some health facilities, primarily due to insufficient storage practices hence the need for a validation survey in 20 health facilities.

### Stage 2- Validation survey in 20 health facilities (Oct. 2022- Mar 2023)

**Completeness of district reporting for PAC data indicators.** Upon analysing the completeness of report submissions across individual districts, we found Mbarara City achieved 100% while Gulu City had a submission rate of 92%. The calculation of submission rates was based on records authenticated at both the district level and by the HMIS focal person at the district headquarters (Fig 2).

**Table 2. Summary of PAC data quality metrics assessed in 20 facilities and their definitions.**

| Post-abortion care (PAC) data quality metrics and data sources reviewed | | | |
|---|---|---|---|
| **PAC data quality metric** | **Definition of the indicator and the WHO recommendation** | **Source(s)** | **Facilities** |
| **Data quality dimension 1: Completeness and timeliness of PAC data** | | | |
| *Completeness of facility reporting in DHIS2* | | | |
| The extent to which health facilities in the district submitted monthly summary reports containing PAC data to the district HMIS focal person | Total number of health facilities expected to submit monthly reports with PAC data versus actual number of health facilities submitting reports with PAC data<br>*WHO recommends 75% or higher reporting* | DHIS2 records extracted at district level | 6 Health Centre II<br>8 Health Centre IIIs<br>4 Health Centre IV and<br>2 Regional Referral Hospitals |
| *Timeliness of facility reporting PAC data in DHIS2* | | | |
| The extent to which health facilities in the district submitted monthly summary reports containing PAC data before the specified timeline | Number and percentage of health facility reports actually submitted on time<br>*WHO has no specific guidance on the timeliness of facility reporting* | Records from the District HMIS focal person | 6 Health Centre II<br>8 Health Centre IIIs<br>4 Health Centre IV and<br>2 Regional Referral Hospitals |
| *Completeness of Indicator data in DHIS2* | | | |
| Extent to which select PAC indicator data within submitted reports contained a non-missing (or non-zero) value. | Number and percentage of non-missing values for a given PAC indicator in expected monthly reports<br>There should be no missing values for a given indicator in 90% or more monthly reports | Records from the District HMIS focal person | 6 Health Centre II<br>8 Health Centre IIIs<br>4 Health Centre IV and<br>2 Regional Referral Hospitals |
| **Data quality dimension 2: Internal consistency of PAC data** | | | |
| *Consistency over time* | | | |
| The extent to which PAC indicator data exhibits similar patterns as in previous years | Ratio of the value of the indicator for the reference year to the end of the preceding 3 years<br>Ratio of the value of the indicator for the reference year should be within ±33% of the mean of the preceding 3 years | DHIS2 | 6 Health Centre II<br>8 Health Centre IIIs<br>4 Health Centre IV and<br>2 Regional Referral Hospitals |
| Outliers in reference year | | | |
| Extent to which the values reported for a given PAC indicator are extreme and potentially implausible | Number of moderate outliers (±2–3 SD from the mean) and extreme outliers (±3 SD from the mean) of monthly values during the reference year<br>Value of the indicator should be within ±2 SD from the mean | DHIS2 | 6 Health Centre II<br>8 Health Centre IIIs<br>4 Health Centre IV and<br>2 Regional Referral Hospitals |
| Accuracy I: Consistency between related data | | | |
| The extent to which the values for two or more PAC indicators exhibit the predicted relationship | Ratio for values of indicator pairs that have a predictable relationship<br>Indicator pairs that should be roughly equal should be within ±10% of each other. | DHIS2 | 6 Health Centre II<br>8 Health Centre IIIs<br>4 Health Centre IV and<br>2 Regional Referral Hospitals |
| Accuracy II: Consistency between original facility registers and reported data in DHIS2 | | | |
| The extent to which values for given PAC indicators agree between two internal data sources | Ratio of indicator values in original facility register count to facility monthly summary report in DHIS2<br>Indicator value in the original facility register count and facility monthly report in DHIS2 should be within ±10% of each other. | DHIS2 and Facility registers | 6 Health Centre II<br>8 Health Centre IIIs<br>4 Health Centre IV and<br>2 Referral Hospitals |

Notes: PAC = post-abortion care, SD = standard deviation.

*The 75% benchmark applies only to completeness of reporting as recommended by the WHO DQR toolkit. Accuracy I and II are internally derived indicators without a WHO-defined cut off; lower values indicate greater discrepancies between data sources.

**For the four-year period that was studied, downloaded health facility data from Uganda's DHIS2 did not distinguish between missing values and true zero values; both are presented as missing values.

Fig 2 shows that Pallisa (61.3%) and Nakasongola (57.9%) did not meet the World Health Organizations' (WHO) recommended 75% target for how complete their monthly reports should be.

**Completeness and timeliness of facility reporting.** We examined health facilities' data submissions to assess the completeness of reporting. For completeness, we verified whether all health facilities required to send PAC data to the district's main health information officer did so and whether this data was subsequently reflected in the Ministry of Health's DHIS2 system. Completeness of facility reporting was calculated as the total number of health facilities expected to submit monthly reports with PAC data (denominator) versus actual number of health facilities submitting reports with PAC data (numerator). According to WHO cut-off of 75%, if a district had less than 75% of the facilities not reporting, on average, that district was considered to be low reporting (Fig 3).

From the health facilities randomly sampled, on average, the overall completeness rate was above 90% every month, as seen in Fig 3. Additionally, when it comes to sending monthly reports with Post-Abortion Care (PAC) data, all the selected health facilities consistently submitted their reports by the 7th of each month achieving a 100% rate of timely submissions.

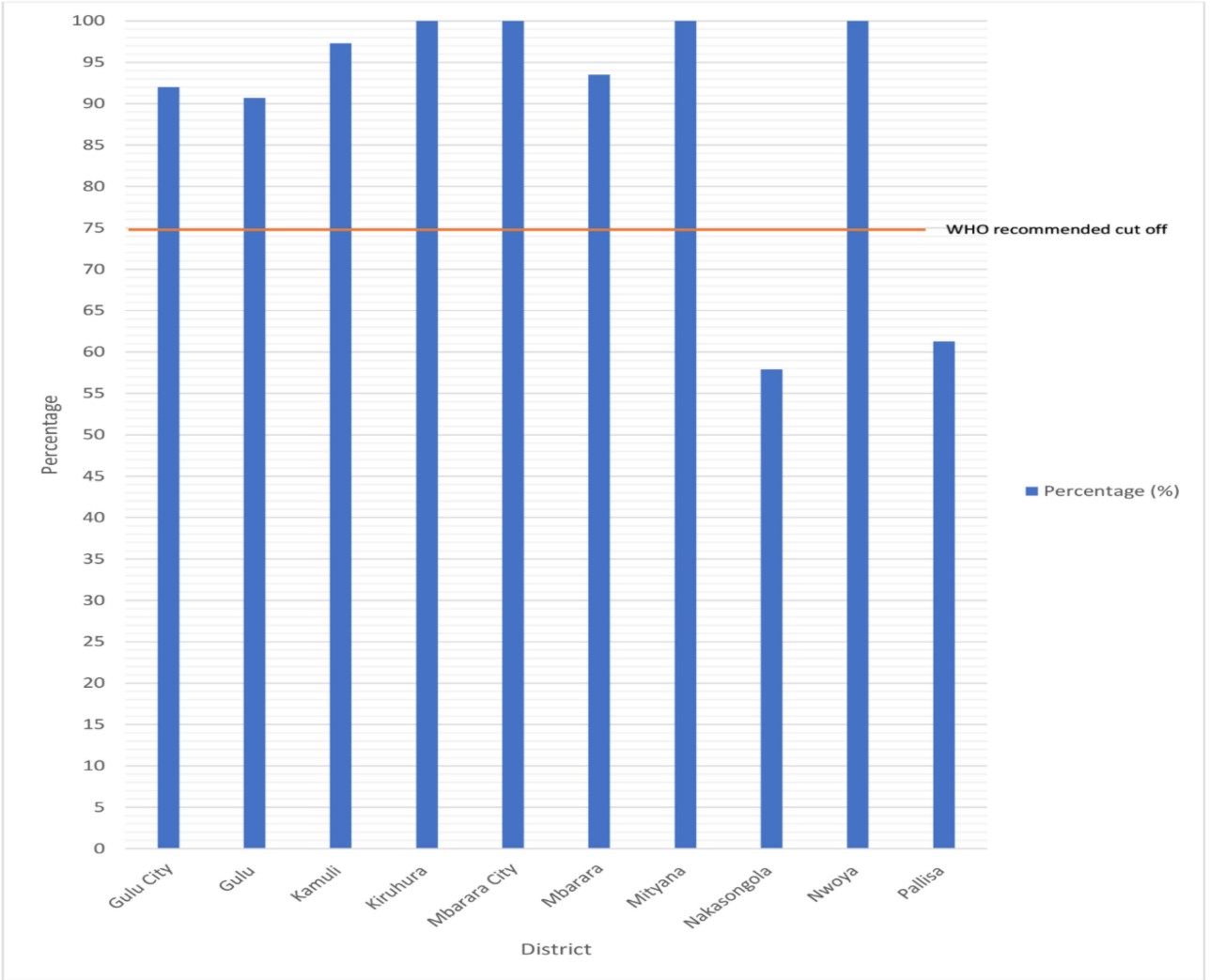

**Fig 2. Average completeness reporting by district.**

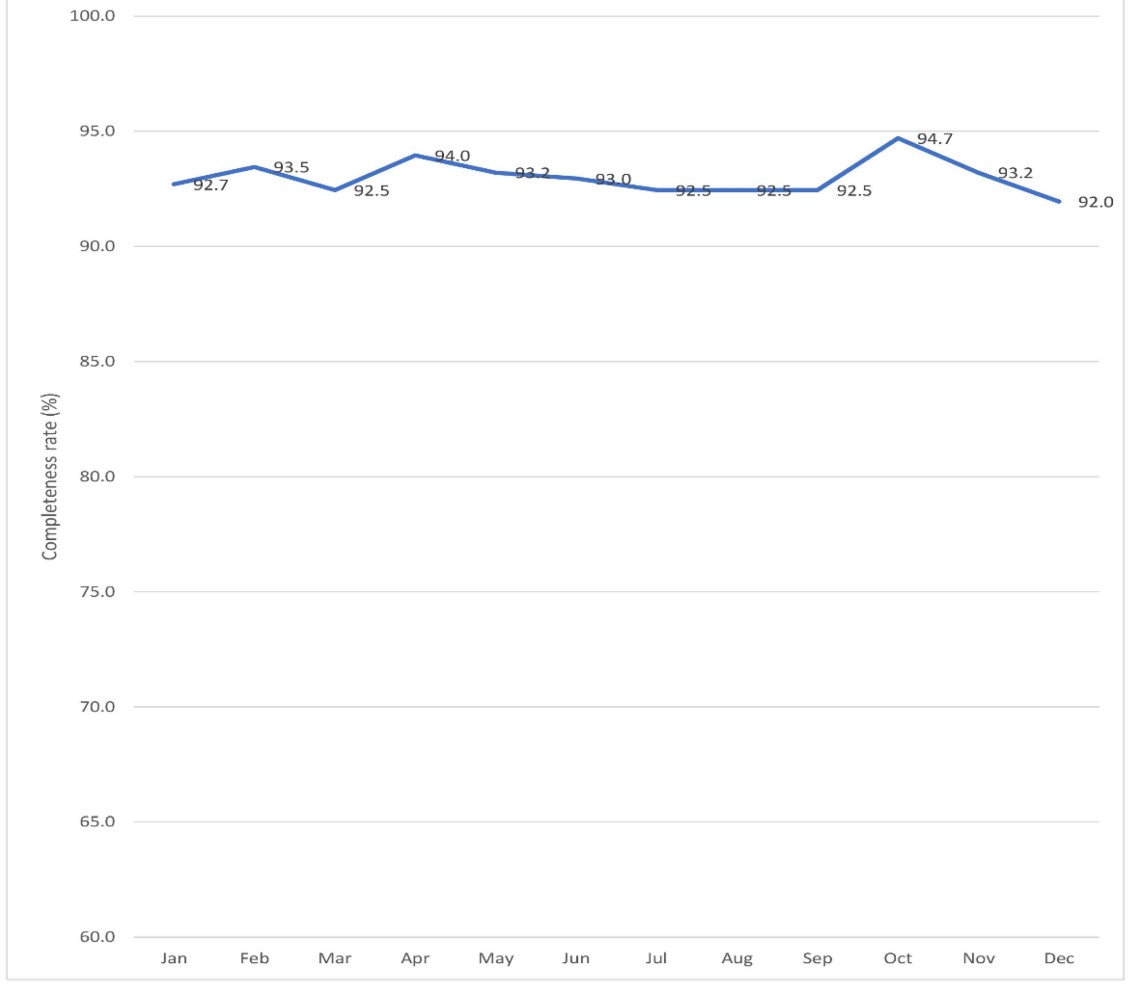

**Fig 3. Overall trends in completeness of reporting by sampled health facilities.**

**Timeliness of facility reporting.** Timeliness of facility reporting. For timeliness, we monitored if these facilities submitted their reports by the 7th of each month and if districts forwarded this data to the Ministry of Health by the 15th of the same month.

## Consistency over time

This metric highlights the stability of crucial maternal indicators for 2021 against the preceding three years. The use of post-abortion family planning (PAFP) as the base measure for post-abortion care is informed by the Essential Maternal and Newborn Clinical Care guidelines for Uganda which mandates that women receiving post-abortion care should also get family planning services [22].

Table 3 shows that for the first year (January 2018 to December 2018) and the second year (January 2019 to December 2019), there were no records for PAFP and incomplete evacuations.

A consistency check for other maternal indicators revealed discrepancies. The figures for abortions due to gender-based violence (1.96), abortions for other reasons (1.71), the first antenatal care ANC) visit for women (1.62), the fourth ANC

Table 3. Consistency of reporting maternal indicators over time.

| | Year 1, Jan 2018-Dec 2018 | Year 2, Jan 2019-Dec 2019 | Year 3, Jan 2020-Dec 2020 | Year 4, Jan 2021-Dec 2021 | Mean of Year 1-Year 3 | Consistency (Ratio of Year 4 to Mean of Year 1-Year 3) |
|---|---|---|---|---|---|---|
| **Main denominator** | | | | | | |
| PAFP | 0 | 0 | 205 | 438 | – | – |
| **Abortions** | | | | | | |
| GBV | 16 | 16 | 37 | 45 | 23 | **1.96** |
| Other causes | 1487 | 1299 | 4097 | 3918 | 2294.33 | **1.71** |
| Incomplete evacuations | 0 | 0 | 1019 | 1200 | – | – |
| ANC 1st visit for women | 41421 | 44704 | 48719 | 73031 | 44948 | **1.62** |
| ANC 4th visit for women | 17271 | 21824 | 26673 | 33205 | 21923 | **1.51** |
| ANC 4 + visits for women | 5126 | 10237 | 18976 | 20461 | 11446 | **1.79** |
| Total ANC contacts/visits (new clients + re-attendances) | 117941 | 138457 | 161532 | 175926 | 139310 | 1.26 |
| Admissions | 33637 | 38231 | 43580 | 50751 | 38483 | 1.32 |
| Deliveries in unit – Total | 26466 | 29532 | 34398 | 40767 | 30132 | 1.35 |
| Deliveries in unit – Live births | 25976 | 29103 | 27264 | 36630 | 27448 | 1.33 |
| Deliveries in unit – Fresh stillbirth – Total | 242 | 252 | 281 | 289 | 258 | 1.12 |
| Deliveries in unit – Macerated stillbirth – Total | 288 | 272 | 283 | 304 | 281 | 1.08 |
| Malaria in pregnancy (cases) | 1746 | 3155 | 3257 | 2491 | 2719 | 0.92 |
| High blood pressure in pregnancy (cases) | 188 | 262 | 318 | 228 | 256 | 0.89 |

*According to WHO guidance, ratios <0.67 or >1.33 indicate reported data in DHIS2 for the reference year was inconsistent with the mean of the preceding 3 years.

visit for women (1.51), and the ratio for women attending more than four ANC visits (1.79) all exceeded the World Health Organisation's recommended threshold of 1.33 for consistent data. This suggests that the data reported in DHIS2 for the year 2021 were notably higher than the average of the previous three years for these specific maternal health indicators.

**Outliers in the reference year.** The integrity of the data sent to the DHIS2 was verified by identifying any outliers and recording their extent using the MAD method, applying the Hampel X84 rule, which detects outliers in non-normally distributed data. Values exceeding ± 5 MAD from the district-or facility-type annual median were flagged as outliers, consistent with WHO recommendations for assessing data consistency.

Analysis conducted based on the type of health facility As shown in Table 4 indicated that Health Centers II and III exhibited a significantly higher number of outliers across most health indicators than Hospitals and health Center IVs. Notably, post-abortion care (PAC) had a substantial number of outliers in these lower-level facilities, suggesting potential challenges with the accuracy of cadre documentation. Similarly, high outliers were observed in live births and admissions

Table 4. Case number of outliers by health facility level.

| Facility Level | ANC 1 | ANC 4 | ANC total | Admissions | Live births | PAC | Post-abortion FP | Freq. (N) | % |
|---|---|---|---|---|---|---|---|---|---|
| Hospital | 0 | 0 | 0 | 0 | 0 | 0 | 24 | 240 | 6.1 |
| HC IV | 1 | 1 | 0 | 0 | 0 | 2 | 9 | 192 | 4.8 |
| HC III | 1 | 12 | 3 | 3 | 9 | 637 | 35 | 1,896 | 47.4 |
| HC II | 31 | 70 | 43 | 799 | 726 | 412 | 22 | 1,668 | 42 |

at Health Centers II and III, raising concerns about data reliability from these facilities. In contrast, hospitals demonstrated better reporting accuracy, particularly in antenatal care and maternity admissions. These discrepancies in data recording at lower-level health centers suggest that Health Centers II and III may face challenges in consistently documenting key health indicators related to maternal and reproductive care.

A further review of data at district level showed that the PAC indicator presented the highest number of outliers. Among all indicators evaluated, Mityana district reported the highest frequency of outliers, whereas Mbarara district had the fewest, as detailed in Table 5

**Accuracy I: Consistency between related PAC indicators.** Additionally, the coherence of the reported data was evaluated by examining the relationship between the number of women receiving post-abortion care and those obtaining post-abortion family planning services. The analysis also explored whether an association existed between the number of women attending their first antenatal care (ANC) visit and those completing four or more visits. The results are summarized in Table 6.

PAFP uptake remained low across all types of PAC services in 2020 and 2021. For outpatient PAC cases, the ratio was 0.13 in 2020 and 0.28 in 2021, with a weak but statistically significant positive correlation ($r = 0.113$, $p < 0.001$). For inpatient PAC cases, the ratio increased slightly from 0.06 to 0.12, also showing a weak positive correlation ($r = 0.064$, $p = 0.004$). When considering all PAC cases, the ratio was 0.04 in 2020 and 0.08 in 2021, with a similarly weak but significant correlation ($r = 0.083$, $p < 0.001$). Regarding ANC, the ratio of women attending only one ANC visit compared to those completing four or more declined from 8.08 in 2018 to 3.57 in 2021. This downward trend suggests a gradual improvement in ANC follow-up. A moderate positive correlation was found ($r = 0.236$, $p = 0.001$), indicating increasing consistency in ANC service utilization over time.

**Table 5. Case number of outliers by district for 5 STD from the median.**

| District | ANC 1 | ANC 4 | ANC total | Admissions | Live births | PAC | Post-abortion FP | Freq. |
|---|---|---|---|---|---|---|---|---|
| Gulu City | 20 | 31 | 26 | 48 | 47 | 51 | 7 | 192 |
| Gulu District | 2 | 9 | 10 | 57 | 52 | 146 | 3 | 288 |
| Kamuli District | 0 | 1 | 0 | 0 | 0 | 128 | 13 | 432 |
| Kiruhura District | 0 | 0 | 1 | 0 | 0 | 117 | 3 | 480 |
| Mbarara City | 0 | 4 | 0 | 120 | 118 | 37 | 5 | 288 |
| Mbarara District | 12 | 2 | 0 | 7 | 0 | 65 | 1 | 432 |
| Mityana District | 46 | 42 | 46 | 48 | 35 | 239 | 32 | 480 |
| Nakasongola District | 18 | 11 | 36 | 56 | 52 | 186 | 9 | 480 |
| Nwoya District | 18 | 46 | 38 | 107 | 89 | 230 | 12 | 468 |
| Pallisa District | 0 | 4 | 1 | 48 | 43 | 217 | 5 | 456 |

**Table 6. Annual ratios and correlation between post-abortion family planning and antenatal care indicators.**

| Predictor Variable | Ratio* | | | | Pearson Correlation** | *P*-Value*** |
|---|---|---|---|---|---|---|
| | **2018** | | | **2019** | **2020** | **2021** |
| Post-abortion family planning vs outpatient PAC cases | – | – | 0.13 | 0.28 | 0.113 | 0.00 |
| Post-abortion family planning vs inpatient PAC cases | – | – | 0.06 | 0.12 | 0.064 | 0.004 |
| Post-abortion family planning vs all PAC cases | – | – | 0.04 | 0.08 | 0.083 | 0.00 |
| ANC first visit vs ANC 4 + visits for women | 8.08 | 4.37 | 2.57 | 3.57 | 0.236 | 0.001 |

*Ratios compare post-abortion FP uptake to PAC cases and ANC 1st to 4 + visits.,**Pearson correlation measures the relationship over time,***p-values show significance (p < 0.05), (–) indicates missing numerator data.

## Accuracy II: Consistency between facility register data and DHIS2 reports

To assess the accuracy of reported data, we compared values for selected indicators documented in the original facility registers with corresponding values reported in the DHIS2 system multiplied by 100. An accuracy score of 100% indicated perfect agreement between the two sources. Scores greater than 100% indicated underreporting of health facility data in the DHIS2 system, while scores less than 100% indicated overreporting of health facility data in the DHIS2 system. While the metric is based on WHO's accuracy II criterion, visual time-series plots were added to illustrate discrepancies across months. As such, no trend consistency scores (e.g. slope or correlation of paired indicators) were computed for this comparison (Fig 4).

Fig 4 illustrates patterns of overreporting in antenatal care (ANC) data submitted to the DHIS2 system, with notable discrepancies in total ANC visits, especially during October 2021. This was due to high rates of data entry errors, such as double counting and incorrect aggregation of visit data, leading to inflated numbers for total ANC visits.

Fig 5 shows a similar pattern of overreporting of total admissions and live birth data in the DHIS2 system. Overreporting occurred most frequently in November 2021.

Fig 6 shows that PAC data is the least accurate of the maternal health indicators, with over-reporting in the DHIS2.

An in-depth analysis of record accuracy by type of health facility revealed that hospitals and Health Centre IVs maintained more precise PAC records over the years compared to Health Centre IIs and IIs, as shown in Table 7. This highlights the urgent need for training medical staff in accurate record-keeping at these lower-level facilities, which serve a substantial number of women in need of PAC.

## Discussion

We evaluated the quality of PAC data reporting across various health facilities in Uganda. The average completeness rate exceeded 90% monthly, with 100% timeliness in submission. Maternal indicators like abortion due to GBV and other

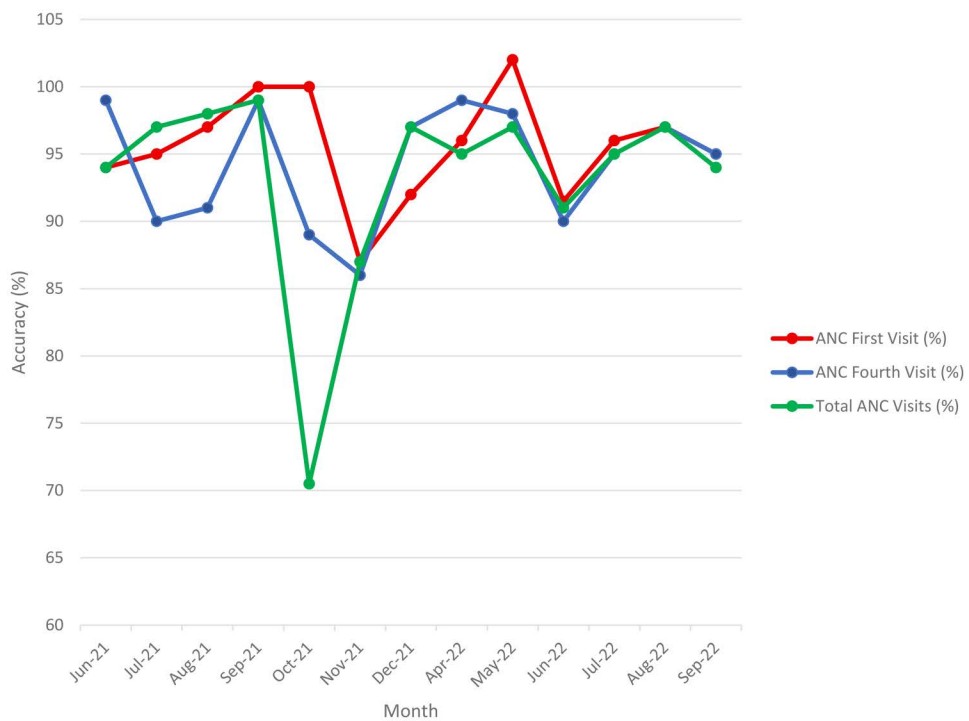

**Fig 4. Accuracy of reporting ANC visits June 2021-September 2022.**

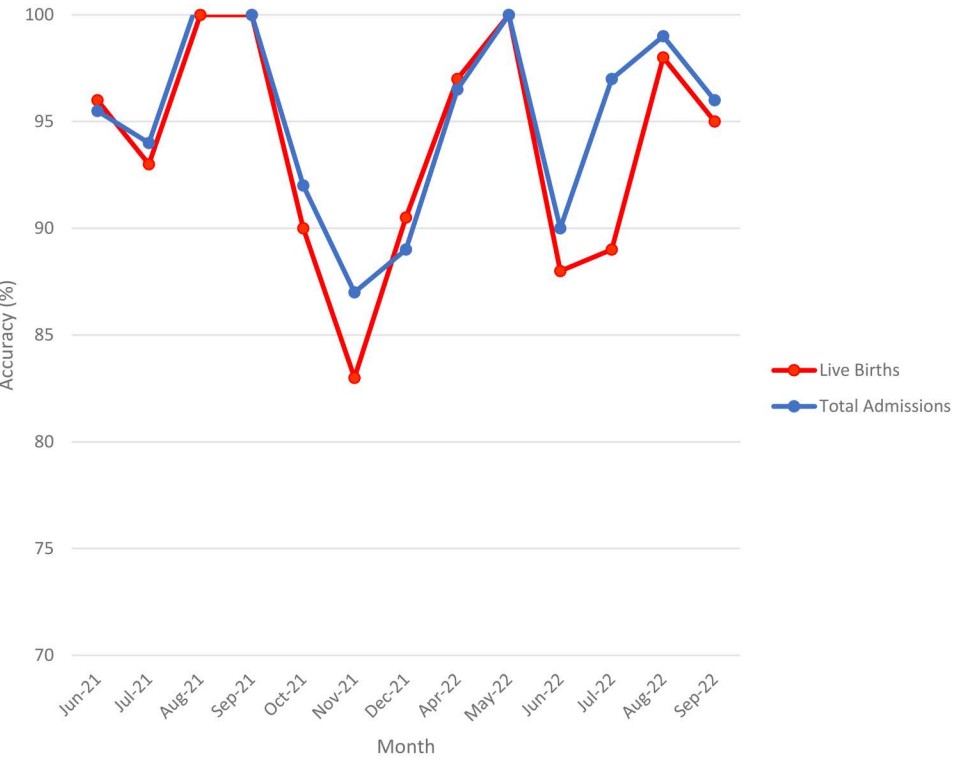

**Fig 5. Accuracy of total admissions and live births June2021-September 2022.**

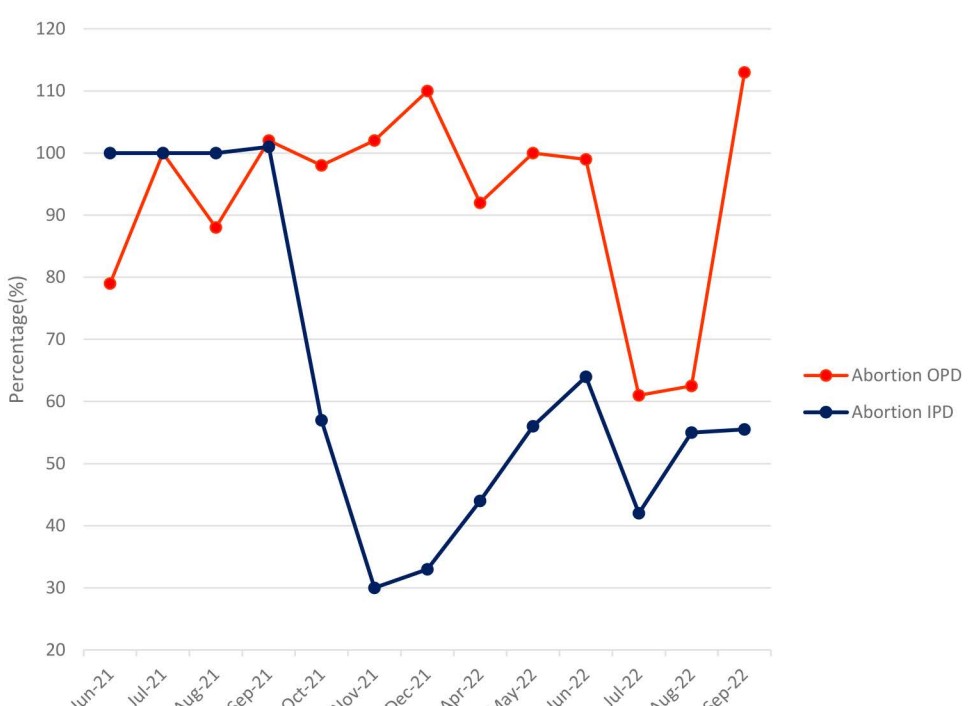

**Fig 6. Accuracy of abortion data June2021-September 2022.**

**Table 7. Accuracy of reporting rate for PAC records by health facility over time.**

| Health Facility Type | 2018 | 2019 | 2020 | 2021 |
|---|---|---|---|---|
| HC II | 8.13 | 6.27 | 12.43 | 20.29 |
| HC III | 10.35 | 16.21 | 29.67 | 21.13 |
| HC IV | 91.35 | 78.69 | 11.13 | 5.34 |
| Hospital | 38.04 | 81.46 | 49.57 | 55.65 |

causes, first ANC visits, and fourth or more ANC visits were all consistently above the WHO recommended threshold of 1.33. Analysis showed that Health Centre IIIs and IIs had more reporting inconsistencies compared to hospitals and Health Centre IVs. The study also found PAC data accuracy less reliable due to missing records and poor storage practices. These findings highlight the need for improved data accuracy and record-keeping in Uganda's health system.

Data are necessary for effective management, accountability and decision-making concerning women who have undergone abortion or face other reproductive maternal challenges. Addressing the obstacles hindering routine PAC data collection and transmission, the United Nations Sustainable Development Goal number three (SDG 3) emphasizes the necessity to enhance health system funding [23]. This would lead to increased human resources for data management and transmission within the health information system, along with improved access to healthcare providers, ultimately resulting in improved individual well-being.

## Completeness and timeliness of facility reporting

The study highlights the strong performance of health facilities in reporting Post-Abortion Care (PAC) data—as part of the overall health facility data, with over 90% of the facilities reporting monthly. This reflects well on the health information systems in place. It aligns with other research emphasizing data integrity and the impact of electronic health records on healthcare delivery and policy-making [24–26]. The success suggests potential best practices for improving data quality in other areas or systems facing similar challenges, highlighting the critical role of complete health data in enhancing healthcare outcomes and informing policy

## Timeliness of facility reporting

The study also found that all visited health facilities achieved 100% timeliness in report submission, indicating exceptional performance in this key dimension of data quality. This result contrasts with existing evidence from low-and middle-income countries, where delayed reporting and data gaps remain common challenges [3,27]. This study demonstrates that complete and timely reporting is achievable, particularly when supported by targeted supervision and digital reporting platforms such as DHIS2. These findings align with global recommendations emphasizing the strengthening of health information systems to support service delivery and informed decision-making.

## Consistency of reported data

The study results also showed that Maternal indicators like abortion due to GBV and other causes, first ANC visits, and fourth or more ANC visits varied but were all consistently above the WHO recommended threshold of 1.33. These results are in tandem with a Kenyan study that evaluated the quality of maternal health data, such as ANC visits and Prevention of Mother-To-Child Transmission (PMTCT). In the study, the authors also found variability in consistency and plausibility across different health indicators, suggesting some data were more reliable than others [28]. This highlights the need for continuous data quality monitoring and evaluation across all health indicators. Health systems should implement regular data quality assessments and provide training to healthcare workers on accurate data recording and reporting practices.

Relatedly, the lower-level health facilities— HCIIs and HC IIIs experienced the highest incidence of outliers, while hospitals and HC IVs recorded the lowest number of outliers. These results mirror a Ghanaian study that assessed MCH data quality in the Cape Coast Metropolis and found that while completeness and consistency of the various maternal indicators were high, consistency varied across facilities as our study found [5]. This may necessitate lower-level health facilities getting additional resources and support to improve their data collection and reporting practices. The support may involve providing more training for staff, better data management tools, and increased supervision and oversight to ensure data accuracy.

## Accuracy I: Related PAC indicators

The study found that the uptake of PAFP remained consistently low across all types of PAC services, with only slight improvements observed over time. While outpatient and inpatient PAC cases both showed weak but statistically significant positive trends in PAFP uptake, the overall levels remained suboptimal. In contrast, ANC service utilization demonstrated a more notable improvement with a gradual decline in the proportion of women attending only one ANC visit compared to those completing four or more. This trend reflects increasing consistency in ANC follow-up and suggests better engagement with routine maternal health services compared to PAFP.These findings are consistent with prior studies that have reported low but gradually improving uptake of family planning services following abortion in LMICs, due in part to service fragmentation, lack of integration and provider-level constraints [3,27,4]. This result adds new evidence of PAFP uptake disaggregated by PAC service type and aligns with global calls to improve integration of contraceptive counseling and provision into emergency reproductive health services. While ANC services show increasing consistency, the persistently low PAFP uptake highlights a critical gap in continuity of care and reinforces the need for targeted interventions within PAC settings to strengthen contraceptive uptake and reduce unmet need.

## Accuracy II: Consistency between facility register data and DHIS2 reports

We compared raw data from health facility records and data retrieved from the DHIS2 because health facility records are the primary data source, maintained directly by healthcare providers responsible for documenting patient care and health services in real time. These records are typically preserved as part of routine clinical and administrative processes, making them less susceptible to data loss or errors that can occur during subsequent reporting stages [14]. Moreover, health facility records are often subject to audits and verification processes, which adds an additional layer of confidence to their accuracy [29].

In contrast, data entry into systems like DHIS2 may involve multiple levels of aggregation, manual input, and potential for errors due to miscommunication, data entry mistakes, or incomplete reporting [30]. Given the structured environment in which health facility records are maintained and the direct involvement of clinical staff, we are confident that these primary records provide a more reliable and accurate reflection of the actual health services delivered, particularly in contexts where routine audits and checks are implemented.

Study results highlighted the accuracy of PAC data compared with other health indicators, emphasizing PAC data as notably less accurate, with a tendency towards over-reporting within the DHIS2 system. This finding is particularly significant as it reflects on the quality of data critical for maternal health services, which are paramount for guiding health policy and program implementation. Comparatively, the literature and reports on health data accuracy often emphasize the general challenges in achieving high fidelity in health reporting systems. Specific indicators such as those related to abortion are sometimes more prone to inaccuracies due to various factors such as the complexity of data collection processes, the sensitivity of the data in question and the level of training and resources available to health workers [31,32]. The challenges identified in accurately capturing PAC data in Uganda echo findings from other regions and health systems that struggle with maintaining high-quality data across different health indicators [33]. The implications are that emphasis on the lower accuracy of PAC data relative to other indicators signals the need for targeted interventions to improve data

collection and reporting practices specifically for PAC services. This could involve enhanced training for healthcare providers on the importance of accurate data entry, the implementation of more user-friendly data capture tools or more rigorous data quality assurance processes.

Moreover, the observed discrepancies in data accuracy across different health facility types and among various health indicators underline the importance of a systematic approach to addressing data quality issues. For health systems, particularly with restrictive abortion laws similar to Uganda, improving the accuracy of PAC data and other health indicators is crucial for informed decision-making, resource allocation and the effective monitoring and evaluation of health interventions aimed at improving maternal health outcomes [32].

### Improving the DHIS2

Currently, over 80 countries around the world use the DHIS2 platform to manage national health information systems, including in many low- and middle-income settings [30]. This data is used for planning and programming at the country level as a means of improving the health outcomes of populations. However, based on our study, it is difficult to say whether there are any in-built safeguards to prevent the entry of extreme values that may affect the quality of DHIS2 data. For example, the study team observed duplicate PAC entries at one referral health facility during the health facility survey. In addition, it was difficult to distinguish between a true zero and a missing value, which has also been reported in other studies [34]. This makes it difficult to conduct data quality assessments using DHIS2 in its current form for some indicators. Improving the DHIS2 platform requires a combination of technical and systemic actions. These include integrating automated data validation checks to flag outliers during data entry, improving differentiation between true zero values and missing data, and enhancing user interfaces for ease of use at lower-level facilities. Additionally, investing in training for health workers, strengthening supervision, and ensuring consistent internet and electricity access could significantly reduce data entry errors and improve reporting accuracy.

### Study strengths and limitations

This study had a major strength in comparing paper-based monthly summaries from health facilities to their electronic DHIS2 versions. However, there was no gold standard data source for abortions on account of stigma and so it was hard to identify the most complete source to compare DHIS2 records against, even though we were looking at PAC, which is technically not induced abortion.

During the review, some health facility records were missing due to poor storage practices. This may have impacted on the comparison that was made between the health facility data and the data in the DHIS2. There is a need for the government of Uganda through the Ministry of Health to improve the storage of health facility records since these can be used as backup when unwanted people like hackers temper with the DHIS2, as it happened in the United Kingdom [35].

Finally, assessing only 80 health facilities out of the 7,631 that provide PAC in Uganda may not have provided a fully representative picture of the overall quality of post-abortion care across all health facilities in Uganda. In addition, even with a stratified sampling approach that we employed, there is a risk that the selected facilities may not have adequately captured the diversity of health facilities across different regions, levels of care, or types of communities served (urban vs rural). This could lead to biased results, especially if certain strata are underrepresented.

### Conclusions

There is minimal data available on abortions and post-abortion care in the Ugandan HMIS, and the quality of data available is poor according to WHO standards. There was inconsistency of data between what was collected at health facility level with data in the DHIS2 system at the Ministry of Health. Despite high submission rates, PAC data accuracy was notably low, highlighting the need for better data management and record-keeping, particularly in lower-level health

facilities. Addressing these disparities is crucial for improving maternal health outcomes, emphasizing the necessity for targeted interventions to enhance data accuracy and reliability within Uganda's health system. We propose that districts ensure that the collected data drives the decision-making, including program design and health budgeting—this way data will be valued.

## Supporting information

**S1. PAC Data_2 set.**
(XLS)

## Acknowledgments

The authors thank the research assistants, health facility in-charges, district technocrats, and the gatekeepers at the Ministry of Health Uganda that facilitated the study. We are grateful to all our respondents that participated in this study.

## Author contributions

**Conceptualization:** Lynn Muhimbuura Atuyambe, Justine N Bukenya, Jesca Nsungwa-Sabiiti, Richard Mugahi, Onikepe Owolabi, Sharon Kim-Gibbons.

**Data curation:** Lynn Muhimbuura Atuyambe, Justine N Bukenya.

**Formal analysis:** Lynn Muhimbuura Atuyambe, Arthur Bagonza.

**Funding acquisition:** Lynn Muhimbuura Atuyambe, Sharon Kim-Gibbons.

**Investigation:** Justine N Bukenya, Samuel Etajak, Jesca Nsungwa-Sabiiti, Richard Mugahi, Paul Mbaka, Onikepe Owolabi, Sharon Kim-Gibbons, Arthur Bagonza.

**Methodology:** Lynn Muhimbuura Atuyambe, Justine N Bukenya, Jesca Nsungwa Sabiiti, Paul Mbaka, Onikepe Owolabi, Sharon Kim-Gibbons, Arthur Bagonza.

**Project administration:** Samuel Etajak.

**Resources:** Jesca Nsungwa-Sabiitia Nsungwa-Sabiiti, Richard Mugahi, Paul Mbaka, Onikepe Owolabi, Sharon Kim-Gibbons.

**Supervision:** Lynn Muhimbuura Atuyambe, Samuel Etajak, Richard Mugahi, Paul Mbaka, Arthur Bagonza.

**Validation:** Lynn Muhimbuura Atuyambe, Samuel Etajak, Richard Mugahi, Paul Mbaka, Sharon Kim-Gibbons, Arthur Bagonza.

**Visualization:** Kristy Friesen.

**Writing – original draft:** Lynn Muhimbuura Atuyambe.

**Writing – review & editing:** Justine N Bukenya, Samuel Etajak, Jesca Nsungwa-Sabiiti, Richard Mugahi, Paul Mbaka, Onikepe Owolabi, Sharon Kim-Gibbons, Kristy Friesen, Arthur Bagonza.

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
