## [Decision Letter · Decision Letter 0]

16 Jun 2025

PONE-D-24-54815Review of the quality of post-abortion care records across 80 Public Health Facilities in Uganda: What are the gaps and how do we improve the data quality?PLOS ONE?

Dear Dr. Atuyambe,

Thank you for submitting your manuscript to PLOS ONE. After careful consideration, we feel that it has merit but does not fully meet PLOS ONE’s publication criteria as it currently stands. Therefore, we invite you to submit a revised version of the manuscript that addresses the points raised during the review process.

We look forward to receiving your revised manuscript.

Kind regards,

Olushayo Oluseun Olu

Academic Editor

PLOS ONE

 [Finding for this work was secured from Vital Strategies Inc. New York USA. How ever the study ended a year a go.].

Natural Earth (public domain): http://www.naturalearthdata

Additional Editor Comments:

Please note that reviewer 1 made his comments directly into the manuscript. Please ensure that you download and address them. Also ensure that you provide a point-by-point response to all the comments ensuring that you indicate the line and page numbers where the changes were made. Thank you

Reviewers' comments:

Reviewer's Responses to Questions

**Comments to the Author**

1. Is the manuscript technically sound, and do the data support the conclusions?

Reviewer #1: Partly

Reviewer #2: Partly

2. Has the statistical analysis been performed appropriately and rigorously?

Reviewer #1: I Don't Know

Reviewer #2: No

3. Have the authors made all data underlying the findings in their manuscript fully available?

Reviewer #1: Yes

Reviewer #2: Yes

4. Is the manuscript presented in an intelligible fashion and written in standard English?

Reviewer #1: Yes

Reviewer #2: No

Reviewer #1: None, for now. I have already gave my comments in the manuscript. The author may check it and address the queries, if wishes. I think that will be enough to enrich the quality of the revised article.

Reviewer #2: General comments

The authors presented a highly relevant but arduous undertaking of collating and evaluating a large body of routine health information to ascertain how well they perform based on standard data quality metrics.

However, there are some areas of ambiguity that could simply be re-presented in the description of the methods, specifically on the sample size and the selected WHO metrics, while the omission of relevant descriptive and inferential statistics in the results may warrant additional analysis. Furthermore, attention to ensuring consistent and harmonious use, presentation and interpretation of many of the data quality metrics, would greatly improve fidelity to the intentions of the paper.

All these would entail a major revision of the manuscript.

Abstract

PAFP should be written in full.

Authors should note that this result (r =0.083, p<0.001) indicate a very weak relationship albeit positive, thus the evidence is insufficient to submit that there is internal consistency between the records of women who received PAC and those who received PAFP.

Methods

Lines 226-230, it is unclear the exact data quality metric the authors meant here as “WHO’s recommended accuracy rate” given that in their Table 2 of data quality metrics, two specifically termed “Accuracy I and II” were listed and none of them have a WHO reference cut-off of 75% (besides completeness of reporting). Please clarify and harmonize.

The description in lines 231-237 appear to indicate that the study was eventually not conducted in 80 facilities but 20. If so, the authors should make it clearer and consider the need to redirect the study title, the description of sample determination, and the information in Table 2 such that all pertains to 20 facilities, not 80. This would ensure that readers are not confused or misled. It is suggested that a flow diagram be presented, and the mentioned level of 'accuracy' that informed the final choice and number of facilities that study results are based, be posed as eligibility criteria. If otherwise, then the report in lines 231-237 of the initial assessment of accuracy for all 80 facilities qualify to be considered as part of the study findings and thus be presented in the Results section rather than as a methodological process.

Ethical approval

Approval for two additional districts was mentioned, however the explanation of their eligibility for selection was not provided in the earlier section where the eight districts that met the high and low caseload categories, were listed.

Results

As mentioned, it needs to be established what number of facilities the results are based on. That said, the convention in health studies is to report both absolute numbers (frequency) along with percentages (proportion); authors should reflect both.

Though it is in order to show results for aggregated data at district level, authors should be mindful that the primary unit of analyses are the facilities. As such, the overall level of completeness, for instance, at facility level should be reported before the comparison between districts and/or between type of facility. This should be the approach to the reporting of the other measures/dimension of data quality. In that regard, authors should merge the on district and facility reporting into one subheading and start with facility-level results.

For the comparison between districts or facility type, the report would be better served by computing statistical tests of significance to allow for statistical inference and generalization of findings, given that only a subset of facilities was selected for each stratum (district or facility type).

In Figure 1 and in the rest of the narrative, what ‘average’ was computed as percentages? This should be stated beforehand under a Data Analysis section, which appears to be missing. Authors should address this by providing that subsection under Methods.

Relating to this, a computation of regression coefficient for internal consistency was stated in the Abstract, again there is no details offered in a Data Analysis section.

Completeness of reporting and timeliness

The description of what was assessed as completeness is not clear and seems different from the WHO criterion. In Lines 279-282, it appears that authors evaluated data completeness by reviewing whether DHIS data were correct with what was sent to the district health information officer. If so, it does not align with the WHO criterion on Table 1, thus should be re-evaluated and reported correctly.

There is another metric for assessing consistency between data sources that appears in authors’ Table 1 as “Accuracy II” – in which case the primary data source is the facility records (not the district officer’s). There is a subsection on Accuracy between original facility data and DHIS2 data (Accuracy II) – see comments below on Figures 3a – 3c.

Authors could consider a separation of the results for completeness and timeliness though it is understandable that the latter has very little information to report.

Consistency over time and Table 3

Lines 310-312: authors should change the term “match” at it communicates a different meaning from what a consistency check attempts to establish. As much as possible, deductive statements should be reserved for the Discussion section.

Internal consistency of reported data by checking for outliers in DHIS2 Data

This subheading should be changed. This metric was labeled “Outliers in reference year” in the methods section, authors should harmonize use of terminologies and reconcile methods used to evaluate any of the metrics with what had been earlier stated as WHO criteria. Thus, authors should be clear on whether they used the module on DHIS-2 platform that performs outlier detection and generates the frequency of occurrence. Otherwise, they should describe any other outlier analysis method that was used.

Again, only aggregates (totals per year, per indicator) at district levels were analyzed (Tables 4 & 5). Authors should prioritize an analysis for all sampled facilities, as the first instance. This demonstrate fidelity to the thrust of the paper, which is a report of data quality performance at facility level.

Consistency of reported data in DHIS2

This subheading should be changed. Again, there is a need to be consistent and harmonize use of terms: this metric was labeled “Accuracy I” which was described as “Consistency between related data the extent to which the values for two or more PAC indicators exhibit the predicted relationship”. Additionally, the WHO method to evaluate it is different from what was shown in the analysis (Table 6). If the WHO method of evaluating and rating any metric was modified, this should be pre-informed in the Methods section.

The authors’ use of the Pearson’s correlation analysis however had some errors. First, Table 6 title suggested it as ‘predicting’ a relationship between the indicators, which is not what this evaluation is about, and besides correlation is not for ‘prediction’ analysis (as against regression). Second, the correlation coefficients were wrongly interpreted as “strong correlation” whereas none fell between 0.5 and 0.9 regardless of having significant P-values.

Accuracy between original facility data and DHIS2 data

This metric was properly labeled but should further be reflected as “Accuracy II” to harmonize the thought process from Methods to Results. In addition, authors should reconcile the method used (appears to be Trend Consistency Analysis) versus what was described as a WHO criterion (Consistency Ratios) in the Methods section. As earlier mentioned, any modification in approach should have been described in the Methods.

Authors stated that records from facilities were presented as percentages of accuracy, but it is not clear how they was actually derived. This should be explained.

That said, the methodology used, which appears like Trend Consistency Analysis, is ideally for testing Accuracy I (consistency between separate but related indicators from one data source) i.e. the previous metric/subheading. This is really what is shown in Figure 3a or 3b where two separate, but related indicators were plotted side-by-side on a line graph.

However, the authors reached too far by comparing the pattern from two separate graphs as basis to analyze data accuracy between two data sources - facility data and DHIS data.

Hence, authors should interpret Figures 3a-c as evaluation of Accuracy I using Trend Consistency Analysis and move that under the previous section. The implication would be that three methods were used to evaluate consistency between related indicators: correlation analysis of means of data indicators; trend consistency analysis, and WHO Ratios (which weren’t computed).

The information in Table 7 is not clear; are they proportions or ratios or averages, and how were they calculated?

Consistency between household survey and DHIS2 data

This entire section could be deleted altogether as there was no information, and more importantly, it was not mentioned as part of the evaluation dimensions.

Discussion

This section should be revised in line with any modifications made to the Results section.

**Do you want your identity to be public for this peer review?** For information about this choice, including consent withdrawal, please see our Privacy Policy

Reviewer #1: No

Reviewer #2: **Yes: ** SEYE BABATUNDE

---

## [Author Response · Author response to Decision Letter 1]

17 Jul 2025

Response to reviewers’ comments

Comment 1: A rebuttal letter that responds to each point raised by the academic editor and reviewer(s). You should upload this letter as a separate file labeled 'Response to Reviewers'.

Response: Dear editorial team, I have renamed and attached the required files for this resubmission.

Comment 2: A marked-up copy of your manuscript that highlights changes made to the original version. You should upload this as a separate file labeled 'Revised Manuscript with Track Changes'.

Response: - A marked-up copy file named “Revised Manuscript with Track Changes’ manuscript is attached and uploaded

Comment 3: An unmarked version of your revised paper without tracked changes. You should upload this as a separate file labeled 'Manuscript'.

Response: Revised unmarked manuscript without track changes named ‘Manuscript’ is attached.

Comment 4: Thank you for stating the following financial disclosure: Funding for this work was secured from Vital Strategies Inc. New York USA. However, the study ended a year ago.

Response: The study was supported by an ‘anonymous donor’ through Vital Strategies Inc., New York, USA.

No specific funding or material support was received from the authors’ academic or institutional affiliations. The study did not receive any grants or internal funds from Makerere University or the Ministry of Health, Uganda.

Response: Thanks for the advice. We have now stated in the manuscript that ‘The funders had no role in study design, data collection and analysis, decision to publish, or preparation of the manuscript’.

Response: Authors Lynn Atuyambe, Justine Bukenya, Samuel Etajak, Onikepe Owolabi, Sharon Kim-Gibbons, Kristy Friesen, and Arthur Bagonza were partially funded by an anonymous donor.”

d) If you did not receive any funding for this study, please state: “The authors received no specific funding for this work.”. Please include your amended statements within your cover letter; we will change the online submission form on your behalf.

Response: Authors Jesca Nsungwa Sabiiti, Richard Mugahi, and Paul Mbaka received no specific funding for this work

Comment 5: When completing the data availability statement of the submission form, you indicated that you will make your data available on acceptance. We strongly recommend all authors decide on a data sharing plan before acceptance, as the process can be lengthy and hold up publication timelines. Please note that, though access restrictions are acceptable now, your entire data will need to be made freely accessible if your manuscript is accepted for publication. This policy applies to all data except where public deposition would breach compliance with the protocol approved by your research ethics board. If you are unable to adhere to our open data policy, please kindly revise your statement to explain your reasoning and we will seek the editor's input on an exemption. Please be assured that, once you have provided your new statement, the assessment of your exemption will not hold up the peer review process

Response: De-identified data at individual and facility levels will be made available at any time it is requested. However, it is important to note that in Uganda, abortion is outlawed. While post abortion care is acceptable, ability to correctly identify the two concepts (abortion and post abortion care) is hard for many law enforcement officers and policymakers in Uganda. Thus, the data underlying this study contains sensitive health facility-level information, such as the number of PAC cases per health facility. This is why we prefer that relative restriction of the data as advised by the Research Ethics Committee at Makerere University School of Public Health and the Uganda National Council for Science and Technology.

Data access requests may be submitted to the School of Public Health Research Ethics Committee via the following email address: sphrecadmin@musph.ac.ug or the corresponding author, and will be considered on a case-by-case basis for researchers who meet criteria for access to confidential data.

Comment 6: Your ethics statement should only appear in the Methods section of your manuscript. If your ethics statement is written in any section besides the Methods, please move it to the Methods section and delete it from any other section. Please ensure that your ethics statement is included in your manuscript, as the ethics statement entered into the online submission form will not be published alongside your manuscript

Response Thank you for the guidance. We have moved the ethics statement exclusively to the methods section of the manuscript and deleted it from all other sections. The revised manuscript now complies with PLOS ONE’s formatting requirements regarding ethics reporting.

Comment 7: If you are unable to obtain permission from the original copyright holder to publish these figures under the CC BY 4.0 license or if the copyright holder’s requirements are incompatible with the CC BY 4.0 license, please either i) remove the figure or ii) supply a replacement figure that complies with the CC BY 4.0 license. Please check copyright information on all replacement figures and update the figure caption with source information. If applicable, please specify in the figure caption text when a figure is similar but not identical to the original image and is therefore for illustrative purposes only.

Response: Thank you for the clarification regarding figure licensing. It will take us a long time to obtain permission to publish the figure under the CC by 4.0 license. As such, given the importance of the data, we have removed the figure from the revised manuscript and updated the in-text references accordingly.

Response to Reviewer 1 comments

Comment 1: Page 3, line 80-82: Please add any global or regional evidence to support this statement

Response: Thank you for the comment. We have revised the sentence on page 3, line 80-82 to include relevant global and regional evidence supporting the statement. The updated text now cites recent studies and global reports highlighting the challenges of data quality in maternal health information systems, particularly in sub-Saharan Africa.

Comment 2: Page 3, Line 97-100 quite same as line 93. I would suggest to modify it to avoid the repetition. Moreover, you can add other kinds of barriers that influence data collection and reporting.

Response: Thank you for this helpful observation. We have revised lines 97-100 to remove the repetition with line 93 and incorporated additional barriers that influence data collection and reporting. This revised text now provides a broader perspective on the structural and contextual challenges affecting routine data systems, particularly at lower-level health facilities.

Comment 3: Page 4, line 103 Any reference?

Response: Thank you for pointing this out. We have now added an appropriate reference to support the statement.

Comment 4: Page 4, line 111 to 112: If conducting this study is your ultimate goal, you should explain how and why the effective data from DHIS2 is crucial for decision-making and budgeting.

Response: Thank you for the comment. We have revised the sentence on page 4, lines 111-112 to clarify how and why effective data from DHIS2 is critical for decision-making and budgeting. The revised text emphasizes the role of high-quality routine data in informing resource allocation and maternal health program planning at national and district levels.

Comment 5: Page 4, line 116-117: Check this sentence for grammatical error.

Response: Thank you for highlighting this. We have reviewed and corrected the grammatical error in the sentence on page 4 lines 116-117 to improve clarity and readability.

Comment 6: Page 21, line 357: Please check Figure 3a's description

Response: Thank you for the observation. We have reviewed and revised the description of figure 3a on page 21, line 357 to ensure it accurately reflects the context of the figure and aligns with the data presented.

Comment 7: Page 21, line 357: Please check the Figure 3a's description

Response: Thank you for the observation. We have reviewed and revised the description of figure 3a on page 21, line 357 to ensure it accurately reflects the context of the figure and aligns with the data presented.

Comment 8: Page 27, line 475: Please check the number of countries on DHIS2's platform.

Response: Thank you for pointing this out. We have verified and updated the number of countries currently using the DHIS2 platform based on the most recent data from the official website (www.dhis2.org). The revised sentence reflects the accurate and up-to-date figure, and it now reads “Currently, over 80 countries around the world use the DHIS2 platform to manage national health information systems, including in many low- and middle-income settings”.

Comment 9: Page 27, line 483 to 484: How? What can be done to improve DHIS2 and overcome the challenges?

Response: Thank you for this inquiry. We have revised the sentence on page 27, lines 483-484 to explain how DHIS2 can be improved and what specific actions can help overcome the identified challenges.

Comment 10: Page 29, line 529 (references): Please ensure that the writers keep the same font throughout the whole article.

Response: Thank you for pointing this out. We have reviewed the manuscript and ensured that the font is consistent throughout, including the reference section.

Response to Reviewer 2 comments

Comment 1: PAFP should be written in full.

Response: Thank you for pointing this out. We have revised the manuscript to spell out PAFP in full upon its first mention as “post-abortion family planning (PAFP)” and have retained the abbreviation for subsequent use throughout the text for clarity and consistency

Comment 2: Authors should note that this result (r =0.083, p<0.001) indicate a very weak relationship albeit positive, thus the evidence is insufficient to submit that there is internal consistency between the records of women who received PAC and those who received PAFP.

Response: Again, thank you for this observation. We agree that while the correlation between the number of women who received post-abortion care (PAC) and those who received post-abortion family planning (PAFP) was statistically significant (r = 0.083, p<0.01), the strength of the relationship is very weak. We have revised the interpretation of this result in the manuscript to reflect that the statistical significance does not imply strong internal consistence between PAC and PAFP records. The text now more cautiously notes the limited strength of association and emphasizes the need for further investigation into data linkage and recording practices. In part, it reads, “Although a statistically significant positive correlation was observed between the number of women who received PAC and those who received post-abortion family planning (PAFP) (r =0.083, p<0.001), the association was very weak suggesting limited internal consistency between the two indicators...”

Comment 3: Methods-Lines 226-230, it is unclear the exact data quality metric the authors meant here as “WHO’s recommended accuracy rate” given that in their Table 2 of data quality metrics, two specifically termed “Accuracy I and II” were listed and none of them have a WHO reference cut-off of 75% (besides completeness of reporting). Please clarify and harmonize.

Response: Thank you for this important clarification. We acknowledge the ambiguity and have revised lines 226-230 to accurately reflect the appropriate WHO-recommended benchmark. Specifically, the 75% threshold refers to the completeness of reporting as per the WHO Data Quality Review (DQR) toolkit, and not to the metrics labelled as Accuracy I and II in table 2.

We have removed the reference to a “WHO-recommended accuracy rate” in this section to avoid confusion and we have harmonized the terminology across the text and Table 2. The accuracy indicators are now clearly described as derived from internal comparisons within the data set, without attributing them to a WHO benchmark.

Comment 4: The description in lines 231-237 appears to indicate that the study was eventually not conducted in 80 facilities, but in 20. If so, the authors should make it clearer and consider the need to redirect the study title, the description of sample determination, and the information in Table 2 such that all pertains to 20 facilities, not 80. This would ensure that readers are not confused or misled. It is suggested that a flow diagram be presented, and the mentioned level of 'accuracy' that informed the final choice and number of facilities that study results are based, be posed as eligibility criteria. If otherwise, then the report in lines 231-237 of the initial assessment of accuracy for all 80 facilities qualify to be considered as part of the study findings and thus be presented in the Results section rather than as a methodological process.

Response: Thank you and we appreciate your helpful comment. We confirm that although 80 health facilities were initially sampled and reviewed for completeness and consistency, only 20 facilities met the eligibility criteria (> 50% discrepancy or missing data) and were therefore included in the final analysis. These 20 were selected for an in-depth audit based on data quality concerns identified during the initial desk review. We have revised the title, abstract, sample, sample size description, table 2 caption, and methods section to clearly state that the final analysis was based on these 20 eligible facilities. We have also added a flow diagram to show the selection process and clarified that the study’s findings are limited to these 20 facilities and not representative of the full sample.

Comment 5: Ethical approval - Approval for two additional districts was mentioned, however the explanation of their eligibility for selection was not provided in the earlier section where the eight districts that met the high and low caseload categories, were listed.

Response: Thank you for this important observation. We would like to clarify that the two additional areas—Gulu city and Mbarara city were originally part of Gulu district and Mbarara district, respectively. However, during the period of this study, both cities had been granted anv independent city status, thus were no longer administratively under their former districts (Gulu and Mbarara). As a result, it became necessary to obtain separate ethical clearance for these two city jurisdictions, even though their health facilities were initially considered under the original eight-district sampling frame.

We have revised the methods section to clearly explain this administrative change and the resulting need for additional ethical approvals.

Comment 6: Results - As mentioned, it needs to be established what number of facilities the results are based on. That said, the convention in health studies is to report both absolute numbers (frequency) along with percentages (proportion); authors should reflect both.

Response: Thank you for this comment. Our sample consisted of 80 health facilities randomly chosen across four regions to ensure geographical diversity and stratified by caseload to capture contrasting report contexts (high vs low PAC volumes). The selection of 20 facilities in stage 2 for in-depth validation was guided by predefined criteria of data quality concern (i.e. facilities with > 50% discrepancies or missing data).

Comment 7: For the comparison between districts or facility type, the report would be better served by computing statistical tests of significance to a

---

## [Editor Report · Decision Letter 1]

22 Jul 2025

Assessment of the Quality of  post-abortion care records in 20 Public Health Facilities in Uganda: What are the gaps and how can we improve quality?

PONE-D-24-54815R1

Dear Dr. Atuyambe,

We’re pleased to inform you that your manuscript has been judged scientifically suitable for publication and will be formally accepted for publication once it meets all outstanding technical requirements.

Kind regards,

Olushayo Oluseun Olu

Academic Editor

PLOS ONE

Additional Editor Comments (optional):

Thanks for comprehensively addressing all the comments of the editor and two reviewers. I would suggest that you move the ethics statement to the last sub-section of the methods section before publication.
---

## [Editor Report · Acceptance letter]

PONE-D-24-54815R1

PLOS ONE

Dear Dr. Atuyambe,

I'm pleased to inform you that your manuscript has been deemed suitable for publication in PLOS ONE. Congratulations! Your manuscript is now being handed over to our production team.

Kind regards,

on behalf of

Dr. Olushayo Oluseun Olu

Academic Editor

PLOS ONE